# Snack Food Consumption across the Pune Transect in India: A Comparison of Dietary Behaviors Based on Consumer Characteristics and Locations

**DOI:** 10.3390/nu13124325

**Published:** 2021-11-30

**Authors:** Devesh Roy, Ruchira Boss, Sunil Saroj, Bhushana Karandikar, Mamata Pradhan, Himanshi Pandey

**Affiliations:** 1International Food Policy Research Institute (IFPRI), Agriculture for Nutrition and Health (A4NH), New Delhi 110012, India; r.boss@cgiar.org; 2International Food Policy Research Institute (IFPRI), South-Asia Office (SAO), New Delhi 110012, India; s.saroj@cgiar.org (S.S.); m.pradhan@cgiar.org (M.P.); 3Independent Researcher, Maharashtra 411002, India; kbhushana@gmail.com; 4Centre for Chronic Disease Control (CCDC), New Delhi 110016, India; himipandey@gmail.com

**Keywords:** snack, nutrient intake, snacking patterns, food choices, Body Mass Index, national survey, Pune transect

## Abstract

This study examines patterns of snack food consumption (SFC) in the rural-urban-slum transect (RUST) of a large city Pune and its precincts (population 10 million) in India. The transect structure aims to mimic a representative survey for the location capturing differences by age, gender, urbanicity, and socio-economic levels. Dietary data from 1405 individuals were used to describe snacking patterns and other food consumed at different frequencies; extent of physical activity; and Body Mass Index (BMI) and waist circumference of children, adolescents, and adults. Our results indicate high incidence of SFC across all population age groups, gender, socio-economic levels, and locations. A distinctive finding in relation to studies in high income countries is the prevalence of hunger snacking with 70% identifying hunger as the primary reason for SFC. Apart from hunger, particularly for adolescents, peer influence and social interactions played a significant role in SFC. Dietary behaviors of slum dwellers were characterized by three-quarters of them having SFC together with family members at home. SFC supplemented calories for low-income consumers and complemented calorie intake for high income ones. No significant association with BMI is possibly due to obesogenic SFC being likely offset by lower consumption of non-snack food and higher physical activity among poor and slum dwellers. Promoting awareness about diets and lifestyles, improving physical and economic access to healthier snacks and nutrient dense foods can improve diet quality in a large and heterogeneous population such as Pune.

## 1. Introduction

India still faces the triple burden of malnutrition including underweight, micronutrient deficiencies and rising overweight and obesity [1]. India’s food system is transitioning with significant changes in snacking and eating out behavior across all subpopulations (by age, gender, rural and urban locations, and income classes). Several cities in India particularly in high income or fast-growing states have experienced a brisk expansion of modern food retail and abundance of convenient, cheap, processed, energy-dense, nutrient-poor foods [2].

The Snack Food Consumption (SFC) referred to in this paper includes energy-dense foods often (not always) eaten between meals. Snacks are different from regular meals in terms of nutritional profile, time, and frequency of consumption [3]. Snack foods are energy-dense, nutrient-poor foods which are high in sodium, sugar, and/or fat such as cookies, cakes, sugar-sweetened beverages (SSB), and chips [4]. There is thus a difference between snack and snacking where the latter refers to the act of eating a snack irrespective of its health and nutrition attributes. Several publications designate snacking based on time of consumption or eating occasion [5,6].

More frequent consumption of energy-dense snacks is universally associated with high energy intake and higher proportion of energy in the diet being provided by sugars [7,8]. Increased snacking frequency is associated with a higher odd of overweight and obesity in children and adults [9].

The negative health effects of nutrition transition due to changing diets are compounded by the often low physical activity and sedentary lifestyle, all of which exacerbate the risk of overweight, obesity, and associated noncommunicable diseases. Evidence exists on the nutrient contributions of foods in daily diets by location of consumption where foods eaten outside home had more fat, less fiber, and fewer micronutrients [10]. For the US [11] and for Australia [12], studies have assessed the high frequency of snacking, its constituents, and its contribution to energy intake.

Studies conducted in the UK show a positive association between TV viewing and consumption of fast food and SSB in children as well as negative association with eating fruits and vegetables that is associated with abdominal adiposity in young children [13,14]. There is also variation among young consumers by socio economic status (SES) (in terms of SFC and fruits consumption [15]). Another study conducted [16] and several other studies on SFC cover high-or middle-income countries show similar results [3,17,18,19,20,21,22,23,24,25].

Overall, there are comparatively few studies in developing countries on SFC including in India though studies do show a close association of changing Indian dietary patterns (especially SFC) with obesity and diabetes as compared to traditional diets [26,27]. Some recent studies look at dietary behaviors of Indians [28,29] specifically for SSB [30] and for snacking behavior during the COVID 19 crisis. There is thus a need to study the SFC patterns and their association with nutritional indicators. In a comparable context (in terms of SES), a study conducted in West Java, Indonesia, studies SFC among children with one third of food assessed as SFC [22]. Another study finds snacking prevalence rising but its frequency reducing in China [31]. Other studies looked at the snacking habits in Mexican children which were intermediate, with 68% of 6–13-year-old reporting snacking 1.2 times per day [19].

In India, the latest nationally, regionally, or locally representative dietary data are quite old (in 2011) [32]. Since then, in a decade, there has been a significant food system transformation in India, but its effects could not be analyzed due to data unavailability. With the changes, knowledge of snacking patterns is of special importance to public health policymakers in India with a dual duty of tackling problems related to under- and over-nutrition. The current study tries to fill in the knowledge gap by employing a comprehensive approach and by studying SFC patterns by residence, age groups, SES, and gender in Pune.

As per census 2011, Pune is the ninth largest city by population in India and in the region next only to Mumbai. It is among the set of major IT hubs and manufacturing centers thus having heterogenous and comparatively young residents. The Pune district with 60% urban residents and ~40% in rural areas is rapidly becoming a bustling economic center in the country. Pune has also emerged as a hub of new startups as per Mercer 2017 ‘Quality of Living Rankings’ which evaluates the living conditions across 440 cities and metros in the world. With this economic and demographic growth, the demand for food supplying outlets has been changing. This cross-sectional study was conducted among the urban, slum, and rural populations in Pune covering an array of socio-demographic, economic, nutritional, and health related factors.

The study aimed to (a) examine snacking behavior and determinants of SFC and daily snacking occasions among participants in different age groups and (b) to describe the socio-economic and lifestyle characteristics of individuals in relation to SFC. We recalled SFC that can have obesogenic effects irrespective of time, occasion, and place of preparation. With data from different segments (Pune transect), we try to understand the factors associated with consumption of such obesogenic food. This is also a pilot effort to develop methods for understanding SFC patterns in other locations which have similar characteristics, that is, those of a middle-income city.

## 2. Methods and Materials

### 2.1. Study Population and Design

The study participants were randomly selected from each ward of Pune urban and rural locations. Considering a total population of Pune city of ~3 million, we chose to select a sample of 1470 participants to be interviewed with a margin of error 2% at 95% confidence level. The participants were randomly selected from rural and urban parts of Pune district. Study participants from schools/colleges/organized workplaces/households were selected from villages 50 km away from Pune city in Saswad tehsil (block, a subdistrict unit); villages closer to and far away from the highway were both included.

Urban participants were selected from slums and representative sites distributed in different wards in Pune. To account for attrition and data loss, additional participants were studied from each age group (1405 analyzable/1470 studied). Study information letter explaining activities and benefits of participation in the study was given to all the relevant authorities and participants.

After seeking permission from the authorities, schools and colleges having easy as well as remote access to food (food stalls, street vendors, fast food joints) were selected for studying children and adolescents. In case of adults, both working and non-working participants were interviewed. Non-working adults were interviewed at home while for interviewing working adults, organized workplaces were approached and after seeking permission from the office in charge, those who fit the age group criteria of the study were interviewed. The research team worked with the school/college/office coordinators and enrolled participants, making sure that the study did not disturb routine work of the institution. Among 1405 respondents, 470 were children, 465 were adolescents, and 470 were adults, with equal distribution between men and women based on socio economic status (SES). Among the participants, 53% were from urban areas, 37% from rural areas, and 10% from slums (Figure 1).

An equal number of adolescent male and female were from low and mid SES. Of the female adolescents, 39% belonged to mid SES and 25% to low SES. Among the male adults, 44% belonged to low SES and 36% to mid SES. Children (10 to 12 years), adolescents (16 to 20 years), and adults (25 to 45 years) were randomly selected from schools/colleges/organized workplaces/households. Of the 470 children in the sample, 10% were attending public school, 20% public private school, 30% private school, and 40% were from rural areas; 48% were girls and 52% boys.

Information was collected for age, education, socio-economic levels, household ownership of assets and amenities such as source of lighting, main fuel for cooking, source of drinking water, toilet facility, and durable consumer goods. For each individual component of the Standard of Living Index (SLI, appropriate weights were assigned based on National Family Health Survey 4 (NFHS-4) scoring guidelines, a composite weighted score was computed to derive an index [33]. Focused Group discussions with key informants such as schoolteachers, parents, multipurpose workers, and local vendors were carried out to map access to different snack foods in the study area.

### 2.2. Measurements

The study questionnaire consisted of four modules, including socio-demographic and household status, dietary intake and snack behavior, physical activity, and anthropometry.

Socio-demographic and household status module: Using indicators from NFHS-4, trained research assistants interviewed the study subjects to obtain information on age, education, family size, occupation, marital status, migration status, and religion. Information was collected on socio-economic factors including household ownership of assets and amenities such as source of lighting, main fuel for cooking, source of drinking water, toilet facility, and certain consumer goods. There was a total of 18 questions in this section.

Dietary intake and snack behavior module: Dietary intake assessment was carried out using a food frequency questionnaire (FFQ) [34] with 25 food groups that has been developed for the study with 126 questions. Using this pre-tested FFQ, consumption pattern of routine and snack foods for the past month were recorded. Snack consumption behavior, factors associated with choice of different snacks, and expenditure on them were assessed. For rural households, we also recorded whether the foods grown by them in their farms contributed to their household consumption. Information was collected about the number of times participants had eaten the meals and snacks, their timing, and habits related to skipping meals. The questions asked for responses on intake such as times per day and in the previous 7 days and, given the low-income settings, SFC they had in the previous month, inclusive of beverages.

A variety of two- and three-dimensional food models (bowls, spoons, roti sizes (Bowls, spoons, and roti sizes are taken as measurements to quantify the food portions. The sizes are defined as small, medium, and large. Roti is an Indian bread commonly made from wheat among other grains)) were provided to assist the respondent to estimate the quantity of each food. To obtain an estimate of portion size, subjects were shown common household portions, namely, a glass, cup, small teaspoon, large tablespoon, and a ladle. These were the same portion sizes used at both urban and rural sites. For each food item, the average portion size and the frequency of consumption (per ‘day’, ‘week’, or ‘month’) were documented.

The module further assessed reasons for snacking (hunger, craving, taste, easy availability, access, and company of friends), preference of snacking environment, factors considered while choosing a home prepared and market purchased snack food. Weekly expenditure, label reading habits, meals skipped due to snacking, frequency of eating outside home, and preferred cuisine were also assessed. Incorporating the role of social environment, among the factors behind SFC, questions were asked on snacking with friends, family, and mood (emotional eating). Children and adolescent participants were asked a set of questions including carrying a lunch box from home, pocket money spent on snacks, most common snacks brought home by parents, and snacks available near their school/college. Adolescent and adult participants were also asked if they were following any specific diet.

Physical activity module: We developed a physical activity module with six questions to record time spent in different activities in school/college/office, indoor and outdoor activities, work related physical activity, games and exercise, and mode of transportation. Duration of sleep and sedentary activities such as television watching or video viewing and computer games, tuitions, and homework were also recorded. The duration of time spent and the daily and weekly frequency of each activity were documented separately for weekdays and weekends. Sedentary activity was defined as activities involving an energy expenditure ≤1.5 metabolic equivalents (METs), moderate–intensity activities were defined as 3.0 to 5.9 METs, and activities >6 MET were considered as heavy activities [35].

Anthropometry module: A trained research team performed anthropometric measurements for weight, height, waist, and hip circumference. They were collected in duplicate by the observer with mean values used for the analysis. Height was recorded to the nearest 0.1 cm with the subjects standing erect, head held in the Frankfurt plane, chest at full inspiration, back touching the wall, and shoes removed. Weight was measured to the nearest 0.1 kg on a digital weighing scale (SECA scales; CMS Instruments, London, UK), calibrated with standard weights, and subjects without footwear and with standard indoor clothing. Waist circumference was measured at the mid-point between the lower border of the rib cage and iliac crest, while the hip circumference was measured at the maximum extension of the buttocks using a non-stretchable measuring tape (Ambron). Body Mass Index (BMI) was calculated using the formula weight $\frac{in\;kilogram}{height{\left(in\;meters\right)}^{2}}$. Using cut off points from World Health Organization (WHO) criteria, the subjects were classified into the categories ‘underweight’ (<18.5 kg/$m^{2}$), ‘normal weight’ (>18.5 and <25.0 kg/$m^{2}$), ‘overweight’ (>25 kg/$m^{2}$), or ‘obese’ (>30 kg/$m^{2}$) [36]. In the case of children and adolescents, BMI percentile were calculated as per the CDC and WHO guidelines [37].

### 2.3. Statistical Analysis

The data were analyzed by the STATA software version 17.0 (STATA Corp, College Station, TX, USA). All statistical analyses were performed using the survey data analysis method.

Frequency and weight of monthly SFC and incidence of hunger snacking were calculated based on the percentage of consumers with frequency of SFC and average weight of SFC in per capita terms. Age group, gender, location, and SES were analyzed using the chi-square test (categorial variable) and *t*-tests (continuous variable), respectively with significance defined at *p* < 0.05. Hunger was found significant among reasons for snacking, and the hunger percentage was calculated across the transect.

In the multivariate regression analysis, outcomes selected are monthly SFC frequency, using serving sizes and their weight average monthly consumption (gram/month) of snacks. Covariates included demographic information, objectively measured physical activity data, and motivations for snacking. Demographic information was self-reported and included age group, gender, type of school attended, educational level, household income levels.

Finally, multivariate linear regression was used to examine the association between SFC with age (categorical), sex, urbanicity (rural, urban), and socioeconomic status (SES) based on household assets.

## 3. Results

Average snack consumption was 9.1 items per week, and average monthly SFC frequency was 36.6 times with large variation (standard deviation of 37.5) as shown in Table 1. Monthly SFC frequency monotonically decreases with age, being highest for children at 1.8 snacking occasions per day. Statistically, the differences in snacking frequencies were significant between children and adolescents as well as between adolescents and adults. Strikingly the highest average frequency for SFC was for slum residents at a monthly frequency of 40 occasions.

The weight of consumed snacks was the highest for children (average 2673 g per month) while it was the lowest for adults at 1693 g. The weights of snack food were likely underestimated since no weight measures were found for outside food. Moreover, the weight of snacks consumed in grams was highest in rural areas, being statistically different at a higher level relative to slum and urban locations.

Figure 2 shows SFC frequencies by gender, location, and category of snack comprising 6 groups. The highest frequency is in beverages (including tea). In India, tea is a staple beverage where intake mostly occurs in sweetened form and thus comprises the greatest share of SSB (Figure 2).

In the slums, snacking was comparatively concentrated in fewer items most notably beverages while in rural and urban locations it was more evenly distributed. Adults in slums have relatively lower consumption of sweet snacks, but children in slums had high sugar intake through SFC. There were both qualitative and quantitative differences by gender. Among rural and urban adolescents, girls/women consume more of savory snacks. Across the transect, adults had comparatively low frequency of sweet snacks. Moreover, children across the three locations had a similar frequency of packed snack foods.

SFC was higher in children compared to adolescents and adults on a weekly basis. Tea was consumed daily by 75% adults and 29% adolescents. Consumption of tea on a daily and weekly basis was relatively higher for children in rural areas for both girls and boys compared to urban areas and slums. Tea is often used to kill hunger pangs and is also a baiter beverage for other SFC viz. biscuits, cookies, or some fried snacks. The multiple roles of tea are exemplified in the slum population representing large consumption as SSB [29]. After beverages, sweet snacks and packed snacks are remarkably high for children and adolescents across the whole transect.

Results from the multivariate regression analysis in Table 2 show determinants of SFC. After controlling for co-variates, women have a higher monthly frequency of SFC. Other significant determinants of SFC were migration, parents with literacy, screen time, type of school (government and private), and company of friends.

### 3.1. Reasons for SFC

SFC is comparatively high in urban areas particularly for men but is not increasing secularly with SES, being at highest levels in the slums (Table 1). Hunger is the prime driver of SFC among consumers with low SES. Adolescent males are most likely to report hunger snacking. A higher percentage comprising adolescent males (76%) and adult men (76%) in urban areas and 78% of adult males and 73% of adult females in slums experience hunger snacking.

Table 1 also presents figures for the primacy of hunger as the reason for SFC across subpopulations. Chi square tests (*p* < 0.05) indicate a significantly higher proportion of hunger snacking in urban (including slums) consumers, low SES relative to middle SES, and middle relative to high SES. Beyond hunger snacking, the table in Appendix A provides the full set of reasons for SFC, places for SFC, a checklist for choosing SFC including price and non-price attributes. Expectedly, price (inclusive of discount) is among the important factors in purchase decision of snacks.

Among non-price attributes, the expiry date of the product is somewhat generically important to urban Indian consumers owing to the high incidence of food safety failures and the established culture of looking at expiration date in medicines. Among other nutrition markers, sugar content is the most looked at, though at a small level.

Consumption of fried food on a weekly basis was higher in rural areas especially for adolescent boys (23%). Surprisingly, rural consumption of packed snack and chocolate was higher compared to urban and slums both among children and adolescents. Daily consumption of fried and deep-fried home snacks, outside food and fermented food, was relatively higher for adolescents compared to children and adults.

Hunger snacking and primacy of price in snack choices underline the economic impulse in SFC. Associations of hunger snacking with rural or urban location and with SES were tested and found significant using Pearson’s $\chi^{2}$ statistics (Appendix A).

### 3.2. Place of SFC

The majority consumed snacks while commuting (50.3%) across the transect. Adolescents consumed more snacks with friends, 72% girls and 56% boys in rural areas and 56% girls and 70% boys in urban areas. About 35% of rural men consumed snacks while travelling. Home (61.6%) was the most common place for SFC especially for adult women. About 13% adolescents consumed snacks at college or school. About 27% of rural adolescent boys consumed snacks at mall/hotel/shops and 32% at canteens in urban areas. More than 90% considered price, 66% considered brand, and 64% considered taste while purchasing snacks (highest for rural). Very few ever considered nutrition and food safety while purchasing snacks.

### 3.3. Physical Activity and Screen Time

All participants spent some time (25 min per day) on light physical activities, whereas 97% participants engaged in moderate activity. About 30% participants engaged in heavy activity; excluding children, adults spent more time in heavy activity relative to adolescents. Among children and adolescents, women spent more time performing light activities. The body sizes of children boys and girls including height, weight, and waist were statistically not different, though in the case of adolescents and adults, it was higher for men.

Screen time (ST) was comparatively high averaging 106 min/day; the highest duration being for adolescents at nearly two hours per day. Association of SFC frequency with ST is comparatively high in rural areas. The actual snacking frequency while watching television is likely to be an underestimate as it is inseparable from SFC with family (Table 1) where 80% rural women and 72% in slums consumed snacks with their family.

There was no differential incidence of overweight and obesity (BMI and waist circumference) comparing snack consumers and non-consumers or high and low SFC among children, adults, and adolescents. Higher SFC supplemented calories from other sources among low-income consumers while complementing calorie intake in high SES. Among the poor, the obesogenic element of SFC was also offset by higher physical activity, particularly in the slum.

Table 3 presents the BMI percentile for children and adolescents in the sample. There was no statistical difference based on snacking frequency.

## 4. Discussion

Results show greater likelihood of SFC for women and urban location including slums. Across the Pune transect, adolescent girls consumed more snacks than boys. In comparison to the adults employed in agriculture, unemployed and unmarried adults had higher SFC frequency. Globally, the gender difference in SFC is not unambiguous [39,40,41,42]. In the US, boys had greater SFC [43] with lower SES; sweetened beverage consumption was high, corroborating findings from the study in Mexico, which is in similar economic context, and which showed that for evening snacks, sweetened tea and coffee were the biggest contributors to energy intake in Mexico [19].

The distinctive feature of SFC in the Pune transect is the conspicuousness of hunger snacks. Moreover, possibly reflective of gender bias, more than three quarters female children engage in hunger snacking [44,45,46]. Hedonic binge eating is prevalent among low SES as well.

Parental factors, social environment, and children’s dietary behavior play a role in SFC [47,48,49] where home availability of snacks positively affects SFC (likely consumption of more unhealthy snacks). An important role in dietary behavior is also played by economic factors, including pocket money for both children and adolescents that has been found in other cases. In the Netherlands, studies showed children’s snack-purchasing behavior to be statistically significantly associated with SFC [20]. Children from high SES families were more likely to snack more than children from a lower tertile of SES [19].

School friendships have been shown to be critical to shaping body weight with risk of becoming overweight found higher in children whose friends were overweight [15,20]. Similarly, a positive association has generally been found in the literature between individual and peer SFC: adolescents whose friends consumed many snacks ate more snacks themselves than those whose friends ate few snacks [50]. There are peer effects not only in snacking but also in snack purchasing [51,52]. Similar effects can be seen that are most pronounced in the case of adolescents.

Location may affect food selection for snacks [10,52,53] as well as portion size [54]. Snacking at home or at work might be associated with more healthful snack food choices and eating at other locations with larger snack sizes, higher fat, and lower fiber content [55]. SFC may also be initiated because of celebratory social occasions and availability of or desire for tempting food [4]. Similar associations in terms of snacking at home and celebratory occasions have been assessed here.

Adults with parents who were not illiterate were found to be snacking more. Snacking among children was found to be significant in comparison to adolescents and adults by more than 37% and 93%, respectively. This corroborates studies conducted in other Asian countries where snacking rates among youth (aged 2–19 years) are more variable [56]. Other studies showed high SFC rates among young people, especially school-going children and adolescents [57].

Adolescents with family members migrating were found to be snacking more which was the opposite in the case of children. Respondents engaged in light activities were found to be snacking more than respondents engaged in moderate and heavy activities.

Ordering takeaway food or dining out was significantly associated with SFC frequency for adolescents and adults. Our study observed that a higher proportion of children from government managed school had poor snacking behavior vis-à-vis those from the private school for both demand as well as supply reasons. The survey found many government schools had vendors selling different snacks near the school campus unlike private schools. Our results on lower snacking frequency in private schools (because of countervailing discouragement interventions) contrast with the study done by Mithra et al. [27] who show proportionately higher frequency among students at private colleges (73.6%) than government colleges (55.1%).

Screen time (ST) has been shown to be a significant determinant of dietary behavior including snack choices [50,51,52,53,54,58,59]. Similar results are obtained in our analysis.

### Limitations of the Study

The main limitation of the study was its cross-sectional design, which precludes causal inference. A limitation of the study might also be the use of proxies for completion of the food frequency questionnaire. Moreover, the data collection method for food consumed, which was based on self-reporting, might have caused a bias in reporting. Additionally, self-reported data may be subject to socially desirable answers. As food items’ data were collected via qualitative methods, detailed nutrient intake data were not available. Hence, using only three modes (rarely, weekly, and daily) as qualitative indicators, overlooked cases of consumed food several times and may have under-estimated the results. Moreover, one limitation has been the lack of available weight for outside food.

Even with a detailed set of confounding factors accounted for, it is possible that some confounders may remain such as family characteristics, in addition to unobserved ones such as health awareness and environmental factors. It is recommended that family income and adolescents’ maturity status and their impact be considered as confounding variables in future studies.

## 5. Conclusions

Analysis of the SFC in a large and growing city and its precincts shows the influence of factors operating at the individual, social and environmental levels, consistent with the socio-ecological theory [55]. The current snacking behavior could be explained by the ready availability of snacks, being often more economical than regular food items that disproportionately affect those from lower SES. Another influencing factor could be television viewing. Programs and advertisements on television in turn can promote SFC and enhance poor snacking behavior.

Understanding the contexts and factors of SFC across the transect may assist those involved in the promotion of healthy food habits. Although the dietary guidelines in India mention snack foods, and caution against consuming excessively sweet, savory, or salty snacks, they usually do not provide suggestions for health-promoting alternatives, which is particularly important when a large part of SFC is hunger driven.

At least for the school going children, schools could be the best avenues to reach young children and their families for nutrition education and interventions. Schools can provide a supportive environment in which healthy food choices can be made.

Recommendations to consume more vegetables, fruit, milk, and milk products should dovetail with current snack food preferences. The development of health-promoting snacks could be an important area for collaboration with food enterprises.

### Further Research

One question for future research is to better understand whether any shifts in SFC occur alongside shifts in the food environment which is happening at a fast pace. More work, including experimental studies, might be needed to understand how snacks and eating occasions affect important dietary behaviors (such as portion size or energy density), and how these changes occur across children, adolescents, and adults.

Moreover, the policy response to steer choices towards healthier snacks and nutrient dense foods (taxes/subsidies, awareness campaigns) is an important area for investigation. Assessing the association between ST and dietary habits seems essential for designing and implementing intervention programs for modifiable behavior change in children and adolescents. Moreover, investigations need to be made controlling for dieting status and the weight status of parents when looking at links between SFC and weight status.

## Figures and Tables

**Figure 1 nutrients-13-04325-f001:**
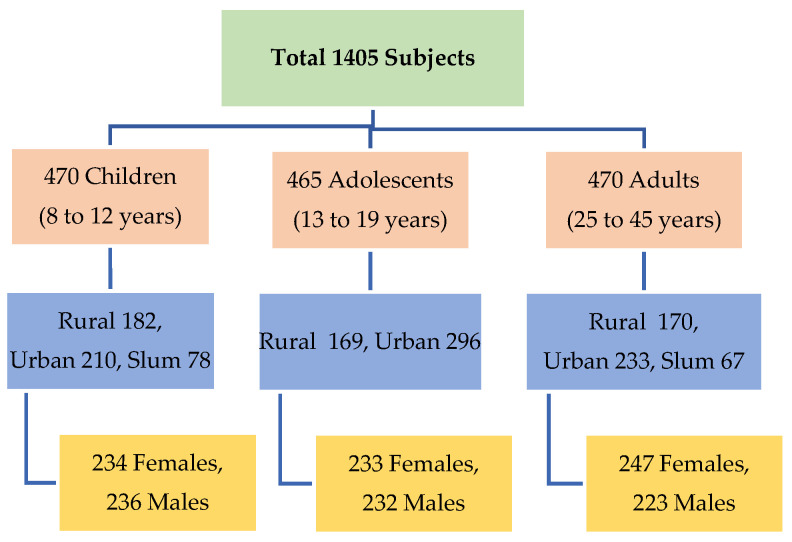
Setting and study participants.

**Figure 2 nutrients-13-04325-f002:**
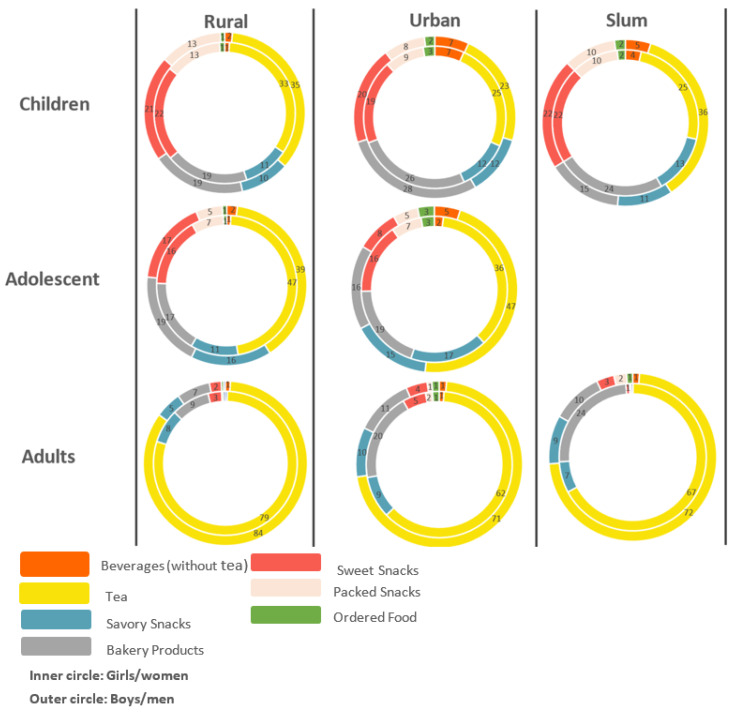
Percentage distribution of monthly frequency of different snacks across groups.

**Table 1 nutrients-13-04325-t001:** Frequency and weight of SFC and incidence of hunger snacking.

	Monthly Freq of SFC		Monthly SFC (in g)		Hunger as a Reason for Snacking (%)		Chi Square for Hunger	
Factors	Mean	SD	Mean	SD	Mean	SD	Pearson Chi Square	Pr
Overall	36.6	37.5	2206.8	3133.0	69.0	46.3		
Children	51.7	43.0	2673.6	3341.6	71.9	45.0	2.86	0.090
Adolescent	36.1	36.1	2254.0	3402.5	66.2	47.3	2.42	0.120
Adult	22.0	25.2	1693.2	2505.7	68.7	46.4	0.02	0.880
*p* for comparison (t value)								
Child and Adolescent	4.5 ***		1.8 *					
Child and Adult	9.5 ***		4.3 ***					
Adolescent and Adult	5.3 ***		2.4 **					
Rural location	34.2	35.8	2736.0	3774.8	62.8	48.4	14.89	0.000
Urban location	37.7	38.2	1810.0	2451.9	73.7	44.0	16.64	0.000
Slum lcation	39.7	39.9	2327.3	3405.9	66.9	47.2	0.32	0.560
*p* for comparison (t value)								
Rural and Urban	−6.2 ***		−6.2 ***					
Rural and Slum	9.9 ***		9.8 ***					
Urban and Slum	15.6 ***		15.5 ***					
Male	33.9	36.3	2081.4	2834.0				
Female	39.2	38.5	2328.1	3394.9				
*p* for comparison (t value)								
Male and Female	2.7 **		1.5 ns					
Low SES	26.6	27.2	1777.9	3170.5	79.8	40.2	35.68	0.000
Mid SES	29.6	32.1	2196.1	2829.4	60.9	48.9	10.60	0.001
High SES	31.4	36.6	1924.5	2976.4	60.4	49.0	8.27	0.004
*p* for comparison (t value)								
Low and Mid	−1.3 ns		−1.3 ns					
Low and High	0.7 ns		0.8 ns					
Mid and High	1.9 ns		1.9 ns					

Note: *** stands for 1%, ** stands for 5%, and * stands for 10% significance level. In the chi-squared test, we used a categorical variable and in the *t*-test, continuous variables were used. ns stands for not significant, Pr for Chi square statistics, SD for Standard Deviation, and g for grams.

**Table 2 nutrients-13-04325-t002:** Determinants of SFC.

OLS Model	Monthly Frequency of Snack Foods (log)						Monthly Consumption of Snack Foods in Gram (log)					
Outcome Variable:	Children		Adolescent		Adult		Children		Adolescent		Adult	
	Coef	SE	Coef	SE	Coef	SE	Coef	SE	Coef	SE	Coef	SE
Hunger ^	−0.025	(0.105)	−0.164	(0.103)	−0.030	(0.136)	−0.051	(0.136)	−0.055	(0.141)	−0.135	(0.160)
Access ^	−0.098	(0.201)	0.765 *	(0.409)	0.038	(0.216)	0.039	(0.243)	0.683 **	(0.281)	0.146	(0.245)
Company of friends ^	−0.000	(0.231)	−0.914 **	(0.414)	−0.334 *	(0.198)	−0.066	(0.265)	−0.777 ***	(0.284)	−0.447 *	(0.232)
Gender (Male = 1; Otherwise = 0)	−0.060	(0.088)	−0.295 ***	(0.093)	−0.205 **	(0.101)	−0.081	(0.100)	−0.223 *	(0.115)	−0.046	(0.114)
Household size (nos)	0.017	(0.021)	−0.011	(0.022)	0.033	(0.023)	0.003	(0.025)	−0.040	(0.027)	−0.007	(0.028)
Illiterate (Mother/ Father = 1; Otherwise = 0)	−0.266	(0.216)	−0.256	(0.219)	−0.244 **	(0.117)	−0.287	(0.195)	−0.412	(0.269)	−0.140	(0.133)
Assets (Base: Poor)												
Middle ^	0.141	(0.107)	0.009	(0.108)	−0.010	(0.121)	0.030	(0.118)	−0.172	(0.127)	0.028	(0.141)
Rich ^	0.070	(0.137)	0.054	(0.143)	0.003	(0.145)	−0.158	(0.157)	−0.296 *	(0.155)	0.200	(0.166)
Location (Base: Rural)												
Urban ^	0.274	(0.203)	0.077	(0.234)	0.503 ***	(0.169)	−0.131	(0.245)	0.291	(0.273)	0.130	(0.194)
Slum ^	0.425 **	(0.198)			0.432 **	(0.191)	0.050	(0.233)			−0.059	(0.214)
Any migrated from family ^	−0.140	(0.134)	0.319 ***	(0.112)	0.127	(0.157)	−0.081	(0.167)	0.136	(0.146)	0.053	(0.223)
Private School Type ^	−0.390 **	(0.161)					−0.268	(0.169)				
Screen time in minutes (log)	0.092 *	(0.056)	0.062 *	(0.037)	−0.002	(0.042)	0.115 **	(0.058)	0.133 **	(0.052)	0.061	(0.053)
Food access based on location (Base: Low)												
Moderate ^	0.407 **	(0.166)	−0.206	(0.200)			0.165	(0.226)	−0.691 ***	(0.250)		
High ^	−0.416 **	(0.207)	−0.184	(0.178)			−0.058	(0.234)	−0.761 ***	(0.219)		
Light Activity in minutes (log)	−0.071	(0.470)	0.230	(0.262)	0.129	(0.172)	−0.143	(0.527)	−0.390	(0.271)	−0.198	(0.191)
Moderate Activity in minutes (log)	0.006	(0.048)	0.069	(0.054)	−0.012	(0.048)	−0.031	(0.057)	−0.021	(0.068)	−0.014	(0.062)
Heavy Activity in minutes (log)			−0.014	(0.024)	−0.025	(0.027)			−0.002	(0.028)	−0.019	(0.033)
Skip Breakfast ^	−0.063	(0.106)	0.116	(0.091)	−0.140	(0.100)	0.016	(0.141)	−0.015	(0.115)	−0.145	(0.119)
Order food ^	−0.122	(0.148)	0.319 ***	(0.107)	0.483 ***	(0.165)	0.066	(0.169)	0.140	(0.151)	0.129	(0.220)
Read label ^	0.013	(0.100)	−0.037	(0.099)	0.060	(0.115)	−0.304 ***	(0.110)	0.074	(0.120)	0.175	(0.134)
Access to snack food (on campus) ^	−0.118	(0.116)	−0.071	(0.137)			−0.086	(0.147)	−0.186	(0.163)		
Access to snack food (nearby campus) ^	0.313 *	(0.184)	0.459	(0.287)			−0.012	(0.239)	0.405	(0.284)		
Carry home tiffin ^	−0.101	(0.198)	−0.127	(0.092)			0.004	(0.228)	0.116	(0.124)		
Checks before buying any snacks												
Price ^	−0.110	(0.140)	−0.237	(0.151)	−0.226	(0.149)	−0.269	(0.168)	0.017	(0.216)	0.042	(0.191)
Brand ^	0.250 ***	(0.089)	−0.068	(0.097)	0.139	(0.113)	0.184 *	(0.100)	0.094	(0.135)	0.049	(0.133)
Taste ^	−0.042	(0.098)	0.153	(0.102)	0.111	(0.119)	−0.028	(0.113)	0.312 ***	(0.120)	0.062	(0.134)
Advertisement ^	0.189 *	(0.107)	−0.177	(0.108)	−0.468 ***	(0.107)	0.133	(0.114)	0.123	(0.124)	−0.481 ***	(0.120)
Availability ^	−0.015	(0.093)	−0.069	(0.091)	0.055	(0.099)	−0.082	(0.123)	−0.013	(0.112)	0.012	(0.115)
Nutrition ^	−0.138	(0.102)	−0.065	(0.092)	−0.206 *	(0.114)	0.060	(0.122)	−0.098	(0.117)	−0.230 *	(0.137)
Food safety ^	0.049	(0.105)	−0.019	(0.094)	−0.190	(0.122)	0.182	(0.130)	−0.077	(0.117)	−0.187	(0.145)
Got Pocket Money ^	0.328 ***	(0.096)	0.236 **	(0.099)			0.337 ***	(0.115)	0.171	(0.118)		
Domestic Help ^	0.346	(0.223)	−0.012	(0.132)	−0.158	(0.205)	−0.120	(0.293)	−0.102	(0.265)	−0.093	(0.254)
Any Disease (Diabetes/Hypertension) ^					−0.148	(0.247)					−0.137	(0.285)
Constant	3.119	(3.174)	1.190	(1.912)	2.030	(1.236)	8.399 **	(3.583)	9.260 ***	(1.997)	8.384 ***	(1.392)
Observations	439		446		450		430		417		418	
R-squared	0.200		0.182		0.226		0.164		0.229		0.084	

Note: *** stands for 1%, ** stands for 5%, and * stands for 10% significance level; ^ denotes binary variable.

**Table 3 nutrients-13-04325-t003:** Distribution of BMI Category/Percentile sample for children and adolescent groups.

Age-Group (0–20 years)		In Numbers			In Percentage (%)		
BMI Category	Percentile	Children	Adolescent	Total	Children	Adolescent	Total
Underweight	<5th	360	130	490	79.3	28.1	53.5
Normal Weight	5th to 85th	80	234	314	17.6	50.7	34.3
Overweight	85th to 95th	11	75	86	2.4	16.2	9.4
Obese	>95th	3	23	26	0.7	5.0	2.8
Total		454	462	916	100.0	100.0	100.0

Source: calculation based on reference developed by Pennington Biomedical Research Center [38].

## Data Availability

Measures derived from the collected data are contained within the article.

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
