# Peer review of "Snack Food Consumption across the Pune Transect in India: A Comparison of Dietary Behaviors Based on Consumer Characteristics and Locations"

_nutrients, 2021, doi:10.3390/nu13124325_

Round 1
Reviewer 1 Report
The manuscript entitled ‘Snack food consumption across the Pune transect in India: A comparison across gender, age, education, socio-economic levels, and locations’ presents interesting issue, however some corrections are needed
- Lines 63-64 – please add the reference to justify it
- Line 69 – ‘Pune’ – why only in Prune? Why it is so important in this region? Please justify it.
- Line 80 – ‘53% participants were’ – A sentence should never start with a number
- More information about methodology must be presented (e.g. about recruitment procedure)
- For the research that involves human subjects the rules of the Declaration of Helsinki of 1975 must be applied (even for on-line survey), including ethics commission approval and especially informed consent. Please add the information about number of ethics commission approval (specific reference)
- Please proper add the information about Informed Consent Statement. Lines 371-373-‘After explaining details of the study to the participants in the local language, adult participants signed an informed consent, adolescents and children signed an assent form. “Taking into account the fact, that the study involvement minors, not only informed consent of minors but also from parents or legal guardians are requited. Please clarify it in the light of local regulations. (please see https://publicationethics.org/news/case-discussion-low-risk-study-no-ethics-
- More information is needed about the validity and reliability of each measure. Additionally, any limitations in reliability and validity need to be addressed in the discussion.
- Line 172 – ‘Error! Reference source not found.’ – please correct it
- Line 175 – ‘(Error! Reference source not found.).’ – please correct it
- Line 193 – ‘Error! Reference source not found.’ – please correct it
- Line 198 – ‘Error! Reference source not found.’ – please correct it
- Lines 186-187 – ‘Tea is often used to kill hunger pangs and is also a baiter beverage for other SFC viz. biscuits, cookies, or some fried snacks’ – it should be expend (please add references)
- Tables should be improved (please see authors guidelines)
- Do not calculate BMI for children and adolescents but calculate BMI percentile (https://www.cdc.gov/healthyweight/bmi/calculator.html) – otherwise it is misleading
- Authors should not reproduce in this section the data that were already presented in previous sections.
- Authors should in their discussion include 3 areas: (1) compare gathered data with the results by other authors, (2) formulate implications of the results of their study and studies by other authors, (3) formulate the future areas which should be studied.
- Authors should present and discuss the limitations of their study.
- Conclusion should be shorten – please focused on the most prominent results
Author Response
Abstract
This study examines patterns of snack food consumption (SFC) in the rural-urban-slum transect (RUST) of a large city Pune and its precincts (population 10 million) in India. The transect structure aims to mimic a representative survey for the location capturing differences by age, gender, urbanicity, and socio-economic levels. Dietary data from 1400 individuals were used to describe snacking patterns, and other food consumed, extent of physical activity, BMI and waist circumference of children, adolescents, and adults. In all, 90% had SFC within a week with higher percentage in urban areas. Distinct from higher income countries, 70% had hunger as primary reason for SFC. Adolescents’ SFC with friends was high showing peer influence>Three-quarters of slum dwellers had SFC with family. There was no association of overweight/obesity (BMI, waist circumference) with SFC or SFC frequency for all age groups. SFC supplemented calories for low-income consumers and complemented for high income ones. Obesogenic SFC was likely offset by lower consumption of non-snack food, higher physical activity among poor and slum dwellers. Promoting awareness about diets and lifestyles, improving physical and economic access to healthier snacks and nutrient dense foods can improve diet quality in a large and heterogeneous population like Pune.
Bottom of Form
Top of Form
Author's Reply to the Review Report (Reviewer 1)
Please provide a point-by-point response to the reviewer’s comments and either enter it in the box below or upload it as a Word/PDF file. Please write down "Please see the attachment." in the box if you only upload an attachment. An example can be found here.
* Author's Notes to Reviewer
FileEditViewInsertFormatToolsTableHelp
Paragraph
P
0 WORDS
Word / PDF
or
Bottom of Form
Top of Form
Review Report Form
Open Review
(x) I would not like to sign my review report
( ) I would like to sign my review report
English language and style
( ) Extensive editing of English language and style required
( ) Moderate English changes required
( ) English language and style are fine/minor spell check required
(x) I don't feel qualified to judge about the English language and style
|
Yes |
Can be improved |
Must be improved |
Not applicable |
|
|
Does the introduction provide sufficient background and include all relevant references? |
( ) |
( ) |
(x) |
( ) |
|
Is the research design appropriate? |
( ) |
( ) |
(x) |
( ) |
|
Are the methods adequately described? |
( ) |
( ) |
(x) |
( ) |
|
Are the results clearly presented? |
( ) |
( ) |
(x) |
( ) |
|
Are the conclusions supported by the results? |
( ) |
( ) |
(x) |
( ) |
Comments and Suggestions for Authors
The manuscript entitled ‘Snack food consumption across the Pune transect in India: A comparison across gender, age, education, socio-economic levels, and locations’ presents interesting issue, however some corrections are needed
- Lines 63-64 – please add the reference to justify it-We have added the reference.
- Line 69 – ‘Pune’ – why only in Prune? Why it is so important in this region? Please justify it.
Response- The city is the seventh largest by population in India and in the region next only to Mumbai. It is among the set of major IT hubs and manufacturing centers thus having a heterogenous and comparatively young residents. Pune district with 60% urban residents and ~40% in rural areas is rapidly transforming to a bustling economic center in the country. The ‘Mercer 2017 Quality of Living Rankings’ evaluated living conditions in more than 440 cities across the world and ranked Pune at 145, second in India after Hyderabad which ranked 144. As per the same ranking, Pune featured amongst evolving business centers and nine emerging cities around the world and was referred as a city which “Hosts IT and automotive companies”. Pune has also emerged as a city of new startup hub. With this economic and demographic growth, the demand of the city for food supplying outlets has increased. This cross-sectional study was conducted among the urban, slum and rural populations in Pune, India, covering an array of socio-demographic, economic, nutritional and health related factors.
- Line 80 – ‘53% participants were’ – A sentence should never start with a number- We corrected
- More information about methodology must be presented (e.g., about recruitment procedure) – In the revision it has been provided. Please see the section revised under Methods and Materials.
- For the research that involves human subjects the rules of the Declaration of Helsinki of 1975 must be applied (even for on-line survey), including ethics commission approval and especially informed consent. Please add the information about number of ethics commission approval (specific reference) – We have provided the ethics committee approval details. The ethics committee approval is provided below.
- Please proper add the information about Informed Consent Statement. Lines 371-373-‘After explaining details of the study to the participants in the local language, adult participants signed an informed consent, adolescents and children signed an assent form. “Taking into account the fact, that the study involvement minors, not only informed consent of minors but also from parents or legal guardians are requited. Please clarify it in the light of local regulations. (please see https://publicationethics.org/news/case-discussion-low-risk-study-no-ethics)
Response- After explaining details of the study to the participants in the local language, adult participants signed an informed consent, adolescents and children signed an assent form. Considering the fact, that the study involved minors, not only informed consent of minors but also from parents or legal guardians was obtained. It was done in accordance with the local regulations.
- More information is needed about the validity and reliability of each measure. Additionally, any limitations in reliability and validity need to be addressed in the discussion. We have added this in the document.
- Line 172 – ‘Error! Reference source not found.’ – please correct it – Format error (delink)- corrected
- Line 175 – ‘(Error! Reference source not found.).’ – please correct it– Format error (delink)- corrected
- Line 193 – ‘Error! Reference source not found.’ – please correct it– Format error (delink)- corrected
- Line 198 – ‘Error! Reference source not found.’ – please correct it– Format error (delink)- corrected
- Lines 186-187 – ‘Tea is often used to kill hunger pangs and is also a baiter beverage for other SFC viz. biscuits, cookies, or some fried snacks’ – it should be expend (please add references)-) This is based on observation based in India and focus group discussions as part of this study. A recent paper on sugar sweetened beverages (SSB) in India that includes tea assesses subjective valuation of tea in energy. It has been referenced (Riediger et al 2021)
- Tables should be improved (please see authors guidelines) –Done
- Do not calculate BMI for children and adolescents but calculate BMI percentile (https://www.cdc.gov/healthyweight/bmi/calculator.html) – otherwise it is misleading--We have provided the BMI percentile for children and adolescents. Please see Table 2
- Authors should not reproduce in this section the data that were already presented in previous sections. We have taken this into account. We have corrected.
- Authors should in their discussion include 3 areas: (1) compare gathered data with the results by other authors, (2) formulate implications of the results of their study and studies by other authors, (3) formulate the future areas which should be studied. Done
- Authors should present and discuss the limitations of their study. We have put in a sub section on limitations of the study
- Conclusion should be shorten – please focused on the most prominent results. It has been shortened
Abstract
This study examines patterns of snack food consumption (SFC) in the rural-urban-slum transect (RUST) of a large city Pune and its precincts (population 10 million) in India. The transect structure aims to mimic a representative survey for the location capturing differences by age, gender, urbanicity, and socio-economic levels. Dietary data from 1400 individuals were used to describe snacking patterns, and other food consumed, extent of physical activity, BMI and waist circumference of children, adolescents, and adults. In all, 90% had SFC within a week with higher percentage in urban areas. Distinct from higher income countries, 70% had hunger as primary reason for SFC. Adolescents’ SFC with friends was high showing peer influence>Three-quarters of slum dwellers had SFC with family. There was no association of overweight/obesity (BMI, waist circumference) with SFC or SFC frequency for all age groups. SFC supplemented calories for low-income consumers and complemented for high income ones. Obesogenic SFC was likely offset by lower consumption of non-snack food, higher physical activity among poor and slum dwellers. Promoting awareness about diets and lifestyles, improving physical and economic access to healthier snacks and nutrient dense foods can improve diet quality in a large and heterogeneous population like Pune.
Bottom of Form
Top of Form
Author's Reply to the Review Report (Reviewer 2)
Please provide a point-by-point response to the reviewer’s comments and either enter it in the box below or upload it as a Word/PDF file. Please write down "Please see the attachment." in the box if you only upload an attachment. An example can be found here.
* Author's Notes to Reviewer
FileEditViewInsertFormatToolsTableHelp
Paragraph
P
0 WORDS
Word / PDF
or
Bottom of Form
Top of Form
Review Report Form
Open Review
(x) I would not like to sign my review report
( ) I would like to sign my review report
English language and style
( ) Extensive editing of English language and style required
( ) Moderate English changes required
( ) English language and style are fine/minor spell check required
(x) I don't feel qualified to judge about the English language and style
|
Yes |
Can be improved |
Must be improved |
Not applicable |
|
|
Does the introduction provide sufficient background and include all relevant references? |
( ) |
(x) |
( ) |
( ) |
|
Is the research design appropriate? |
( ) |
(x) |
( ) |
( ) |
|
Are the methods adequately described? |
( ) |
( ) |
(x) |
( ) |
|
Are the results clearly presented? |
( ) |
( ) |
(x) |
( ) |
|
Are the conclusions supported by the results? |
( ) |
(x) |
( ) |
( ) |
Comments and Suggestions for Authors
The manuscript "Snack food consumption across the Pune transect in India: A comparison across gender, age, education, socioeconomic levels, and locations” by Devesh et al. is quite interesting and covers a topic that was not explored in MDPI journals yet. The study was aimed to learn the snack food consumption, eating behaviors associated with snacking, and anthropometric measurements in a group of 1 400 Indian consumers.
In my opinion, authors should consider shorting the title of this manuscript; there is no point to write all social-demographical aspects measured (they are typical). Maybe you should make it more informative e.g. add information about anthropometrics, and eating behaviors. This is only a kind suggestion for authors, but the final decision is theirs.
Response- Many thanks for the suggestion. We have shortened the title to Snack food consumption across Pune transect: A comparison of dietary behaviors based on consumer characteristics and locations
I think that the authors do not use correct template of Nutrients’ MDPI Journal. Where are the affiliation details and the prospective publication details taken from the left margin? Thank you. We have now provided the details as per the Nutrients’ MDPI Journal template.
This manuscript is full of text editing issues, so it is very hard to focus on more methodological aspects. There are just examples, please check this manuscript thoroughly:
- L5: why “This study” is bolded, why the hyphen is inserted here?-Response- It is corrected
- L 12: “influence>Three-quarter” – what is “>” sign for?- that sign is removed
- L23: please remove this hyphen.-Done
- L42: often-low – please separate it and remove the hyphen.- Done
- L92: “(NHFS)” double bracket should be on the right side.- It is corrected
- L158: “WHO Refs” – probably authors wanted to insert here citation.- corrected
- L172, L175, L193, L197, and elsewhere as appropriate: “Error! Reference source not found” -There was problem due to cross referencing. Now those are corrected
- L210: “Note: ***,** stands for 1%, 5% significance level. Ns-not significant 0”. What is the “0” in the end for? What about one *?-“0” was a typo. It is corrected. We have given a detailed explanation now at the bottom of the table; one * is for 10% significance level
- L310: “Mithra et al (206)” What does 206 mean?- Sorry for the typo. It is supposed to be 2006
I do not feel qualified to correct the English language, but I am sure that some parts of the manuscript require and others would benefit from professional proofreading. Please look at some examples. The text needs to be checked thoroughly: Done
- L31-32: “Snack foods can….” – this sentence is not clear. Require grammar corrections.- We modified the writing for better composition and coherence
- L45-47: “US [10] and [11] for Australia assess the” – not clear. We modified the language
- L48: “Studies conducted on UK, shows a positive”, not clear, should be rather: “Studies conducted in the UK show…”- Corrected
- L49: I suggest inserting “as well as” between “SSB in children” and “negative association”- Done
- L58: “In roughly comparable contexts, a study, in West Java, Indonesia, study SFC” – not clear, at all.- We added comparable context in terms of SES of the study subjects as opposed to the rich countries like UK and Australia
- L64: “In the decade since …” – not clear…- We changed the text to the following. Since then, in a decade, there has been a significant food system transformation in India but the effects of which could not be analyzed due to data unavailability
- L76: “archetypal middle-income city” What does it mean?- We changed to typical or similar
- L94-96: “After seeking…”- not clear- We changed the text to be clearer
- 160: “At weekly frequency, in all, 90% had SFC” – incorrect. We had included tea, coffee (beverages) in the snacking, hence it was showing 90%. We have removed tea now. Instead of weekly we have reported only on monthly frequency now.
- L287: “Results here show” – this is not scientific language- We took out results here show. Thanks for suggesting the correct presentation
- L304: “Ordering outside food” Ordering takeaway? dining out? – not clear- It covers both outside food as well as take away food
- … and many others across the entire manuscript. At some point in this manuscript, the plethora of mistakes caused I stopped finding, correcting both editorial and language issues. This should be done by authors or professional services.- Many thanks for this suggestion. We internally got it checked for grammatical and presentation issues.
Introduction
I general, I think that introduction provides enough background, Nevertheless, please consider my suggestions below:
- L24-28: Please provide a reference- We have provided a reference
- L29: Please provide the full name of SFC in the main text.- We have provided the long form as snack food consumption
- L51: Please provide the full name of SES in the main text. It is a standard scientific practice to provide the full name of the acronym when used for the very first time.- we have provided the full form as socio economic status, the first time the term has been used
- L56: In my opinion, authors should expand this section a bit to provide the newest findings from others studies on Snack Food Consumption in India.
- Bhol A, Sanwalka N, Kapasi TA, Piplodwala SZ, Ansari LM, Katawala FM. Changes in Snacking Patterns during COVID-19 Lockdown in Adults from Mumbai City, India. Current Research in Nutrition and Food Science Journal. 2021 Sep 10;9(3).
- and maybe some others…
Response-We have added papers from recent periods on dietary behaviors in India
Materials and methods
- The subsection “study population and design” should begin with a brief description of study design, i.e. generic information on what was assessed and how many people participated in this study. Then the details can be described. Done
- When referring to SLI and NHFS appropriate references should be added. Done
- Maybe a chart will be more informative to illustrate to the readers how this study was performed. We have added a chart
- The methodology description is missing information on how exactly the interview was performed (did recruit participants started with completing the Food Frequency Questionnaire, then they completed a questionnaire on eating behavior, finally anthropometric measurements? How long does the interview take? Did participants complete the questionnaire by themselves or were answering to designated researchers). We have added the details now
- Each questionnaire description should indicate the number of questions asked. Done
- L103-110: Separate paragraph should be dedicated for statistical analyses. Usually is located at the end of the material and methods section. Done
- L80; L82 and elsewhere as appropriate: Sentences should never begin with numerals, I kindly advise paraphrasing all sentences that use such construction. We have corrected
Results
- The authors collected interesting data in a big sample of consumers, but the presentation manner should be corrected. The scope of an article could be easily split into two valuable articles. Currently, authors try to squeeze everything into one article and this makes it hard to understand and find the most important outcomes. Maybe a cluster analysis would help to find the patterns that the authors refer to in the abstract. Thank you for this valuable suggestion. In future research we will organize our paper based on cluster analysis.
- Table 1. The official symbol International System of Units (SI) for gram is "g" following the numeric value with a space. In my opinion, authors should check this table again is quite complicated and unnecessarily stretched. Removing of 1,5 line spacing in the second row of table 1 and changing “Pearson chi-square” for “χ2” should make the table shorter. We have made changes in the table.
- Authors present the Hunger (%) – where was that explained in methodology? We have now explained in the methodology in the “statistical analyses”
- In my opinion figure, 1 is not legible enough. Maybe authors should consider moving the legend from the left side under the set of circles and expanding them a little bit. We have done that now
- The title of the figure is usually located under the graphs. Authors should refer to the figure in the text, which is not the case. Done
- The subtitle“Statistical Analysis” should be renamed – the statistical analysis subtitle is usually located in the material and methods section. Statistics should be used for all data to evaluate the significance of the results, hence a specific subsection is pointless. We have now included “statistical analyses” as a subtitle in the material and methods section only.
- It is not acceptable to insert a table that is two pages long and does not provide an explanation for it. Moreover, only the most important results should be presented. The legend is incorrect. We have shortened the table and provided only variables of interest
- The authors collected anthropometric data but they did not show any of them and did not find any relationship between BMI and snacking, which is a known factor of obesity development. Thanks for pointing this out. We also find there is no relationship between BMI and snacking as found in other studies but contextually it might be different because of hunger snacking and physical activity.
Discussion
- In my opinion, the discussion is written moderately poorly, as barely tackles aspects that were somehow selected by authors. There is a lack of discussion on titular SCF in India. The probable reason for such a state is a too vast scope of the article. I am really curious about other reviewers' opinions. In the revision we have tried to streamline the discussion section.
Conclusions
- Generic, but I am wondering which of these conclusions are discoveries? We have shortened the conclusion and made it more specific
- No other comments for this section, as is nicely written
- SFC, ST – please provide here a full name. Done
Institutional Review Board Statement – authors can provide the number of resolution if is available-Number is not available. Please see below the Ethics Committee approval letter.
References
- In the manuscript, 50 references were used, but less than 25% of them were published within 5 past years. It shows the need to refresh the references a bit. There are many interesting articles, even from this publisher, which findings support your opinion. We have updated the references
Abstract
- I think the abstract should be also corrected. In my opinion, the first thing you should do is to provide 1-2 short pieces of information on the study context/ background, then what was the aim. After that, the most important measures should be briefly described. Does dietary data describe the physical activity and BMI, as stated here? Then you can report the most important results, but do not forget about some on snack food consumption, which is in the manuscript title. We have reworked the abstract as suggested. Many thanks for suggesting a proper structure for the abstract.
According to journal standards, my review categorizes this manuscript as requiring major revisions. I think the authors might need more time to improve the manuscript, but a chance to do it should be given.
Reviewer keeps fingers crossed and waits for the revised version.
Submission Date
25 October 2021
Date of this review
18 Nov 2021 11:59:46
Bottom of Form
© 1996-2021 MDPI (Basel, Switzerland) unless otherwise stated
Disclaimer Terms and Conditions Privacy Policy
Reviewer 2 Report
The manuscript "Snack food consumption across the Pune transect in India: A comparison across gender, age, education, socioeconomic levels, and locations” by Devesh et al. is quite interesting and covers a topic that was not explored in MDPI journals yet. The study was aimed to learn the snack food consumption, eating behaviors associated with snacking, and anthropometric measurements in a group of 1 400 Indian consumers.
In my opinion, authors should consider shorting the title of this manuscript; there is no point to write all social-demographical aspects measured (they are typical). Maybe you should make it more informative e.g. add information about anthropometrics, and eating behaviors. This is only a kind suggestion for authors, but the final decision is theirs.
I think that the authors do not use correct template of Nutrients’ MDPI Journal. Where are the affiliation details and the prospective publication details taken from the left margin?
This manuscript is full of text editing issues, so it is very hard to focus on more methodological aspects. There are just examples, please check this manuscript thoroughly:
- L5: why “This study” is bolded, why the hyphen is inserted here?
- L 12: “influence>Three-quarter” – what is “>” sign for?
- L23: please remove this hyphen.
- L42: often-low – please separate it and remove the hyphen.
- L92: “(NHFS)” double bracket should be on the right side.
- L158: “WHO Refs” – probably authors wanted to insert here citation.
- L172, L175, L193, L197, and elsewhere as appropriate: “Error! Reference source not found” – no comment needed.
- L210: “Note: ***,** stands for 1%, 5% significance level. Ns-not significant 0”. What is the “0” in the end for? What about one *?
- L310: “Mithra et al (206)” What does 206 mean?
I do not feel qualified to correct the English language, but I am sure that some parts of the manuscript require and others would benefit from professional proofreading. Please look at some examples. The text needs to be checked thoroughly:
- L31-32: “Snack foods can….” – this sentence is not clear. Require grammar corrections.
- L45-47: “US [10] and [11] for Australia assess the” – not clear
- L48: “Studies conducted on UK, shows a positive”, not clear, should be rather: “Studies conducted in the UK show…”
- L49: I suggest inserting “as well as” between “SSB in children” and “negative association”
- L58: “In roughly comparable contexts, a study, in West Java, Indonesia, study SFC” – not clear, at all.
- L64: “In the decade since …” – not clear…
- L76: “archetypal middle-income city” What does it mean?
- L94-96: “After seeking…”- not clear
- 160: “At weekly frequency, in all, 90% had SFC” – incorrect
- L287: “Results here show” – this is not scientific language
- L304: “Ordering outside food” Ordering takeaway? dining out? – not clear
- … and many others across the entire manuscript. At some point in this manuscript, the plethora of mistakes caused I stopped finding, correcting both editorial and language issues. This should be done by authors or professional services.
Introduction
I general, I think that introduction provides enough background, Nevertheless, please consider my suggestions below:
- L24-28: Please provide a reference
- L29: Please provide the full name of SFC in the main text.
- L51: Please provide the full name of SES in the main text. It is a standard scientific practice to provide the full name of the acronym when used for the very first time.
- L56: In my opinion, authors should expand this section a bit to provide the newest findings from others studies on Snack Food Consumption in India.
- Bhol A, Sanwalka N, Kapasi TA, Piplodwala SZ, Ansari LM, Katawala FM. Changes in Snacking Patterns during COVID-19 Lockdown in Adults from Mumbai City, India. Current Research in Nutrition and Food Science Journal. 2021 Sep 10;9(3).
- and maybe some others…
Materials and methods
- The subsection “study population and design” should begin with a brief description of study design, i.e. generic information on what was assessed and how many people participated in this study. Then the details can be described.
- When referring to SLI and NHFS appropriate references should be added.
- Maybe a chart will be more informative to illustrate to the readers how this study was performed.
- The methodology description is missing information on how exactly the interview was performed (did recruit participants started with completing the Food Frequency Questionnaire, then they completed a questionnaire on eating behavior, finally anthropometric measurements? How long does the interview take? Did participants complete the questionnaire by themselves or were answering to designated researchers).
- Each questionnaire description should indicate the number of questions asked.
- L103-110: Separate paragraph should be dedicated for statistical analyses. Usually is located at the end of the material and methods section.
- L80; L82 and elsewhere as appropriate: Sentences should never begin with numerals, I kindly advise paraphrasing all sentences that use such construction.
Results
- The authors collected interesting data in a big sample of consumers, but the presentation manner should be corrected. The scope of an article could be easily split into two valuable articles. Currently, authors try to squeeze everything into one article and this makes it hard to understand and find the most important outcomes. Maybe a cluster analysis would help to find the patterns that the authors refer to in the abstract.
- Table 1. The official symbol International System of Units (SI) for gram is "g" following the numeric value with a space. In my opinion, authors should check this table again is quite complicated and unnecessarily stretched. Removing of 1,5 line spacing in the second row of table 1 and changing “Pearson chi-square” for “χ2” should make the table shorter.
- Authors present the Hunger (%) – where was that explained in methodology?
- In my opinion figure, 1 is not legible enough. Maybe authors should consider moving the legend from the left side under the set of circles and expanding them a little bit.
- The title of the figure is usually located under the graphs. Authors should refer to the figure in the text, which is not the case.
- The subtitle“Statistical Analysis” should be renamed – the statistical analysis subtitle is usually located in the material and methods section. Statistics should be used for all data to evaluate the significance of the results, hence a specific subsection is pointless.
- It is not acceptable to insert a table that is two pages long and does not provide an explanation for it. Moreover, only the most important results should be presented. The legend is incorrect
- The authors collected anthropometric data but they did not show any of them and did not find any relationship between BMI and snacking, which is a known factor of obesity development.
Discussion
- In my opinion, the discussion is written moderately poorly, as barely tackles aspects that were somehow selected by authors. There is a lack of discussion on titular SCF in India. The probable reason for such a state is a too vast scope of the article. I am really curious about other reviewers' opinions.
Conclusions
- Generic, but I am wondering which of these conclusions are discoveries?
- No other comments for this section, as is nicely written
- SFC, ST – please provide here a full name.
Institutional Review Board Statement – authors can provide the number of resolution if is available
References
- In the manuscript, 50 references were used, but less than 25% of them were published within 5 past years. It shows the need to refresh the references a bit. There are many interesting articles, even from this publisher, which findings support your opinion.
Abstract
- I think the abstract should be also corrected. In my opinion, the first thing you should do is to provide 1-2 short pieces of information on the study context/ background, then what was the aim. After that, the most important measures should be briefly described. Does dietary data describe the physical activity and BMI, as stated here? Then you can report the most important results, but do not forget about some on snack food consumption, which is in the manuscript title.
According to journal standards, my review categorizes this manuscript as requiring major revisions. I think the authors might need more time to improve the manuscript, but a chance to do it should be given.
Reviewer keeps fingers crossed and waits for the revised version.
Author Response
Abstract
This study examines patterns of snack food consumption (SFC) in the rural-urban-slum transect (RUST) of a large city Pune and its precincts (population 10 million) in India. The transect structure aims to mimic a representative survey for the location capturing differences by age, gender, urbanicity, and socio-economic levels. Dietary data from 1400 individuals were used to describe snacking patterns, and other food consumed, extent of physical activity, BMI and waist circumference of children, adolescents, and adults. In all, 90% had SFC within a week with higher percentage in urban areas. Distinct from higher income countries, 70% had hunger as primary reason for SFC. Adolescents’ SFC with friends was high showing peer influence>Three-quarters of slum dwellers had SFC with family. There was no association of overweight/obesity (BMI, waist circumference) with SFC or SFC frequency for all age groups. SFC supplemented calories for low-income consumers and complemented for high income ones. Obesogenic SFC was likely offset by lower consumption of non-snack food, higher physical activity among poor and slum dwellers. Promoting awareness about diets and lifestyles, improving physical and economic access to healthier snacks and nutrient dense foods can improve diet quality in a large and heterogeneous population like Pune.
Bottom of Form
Top of Form
Author's Reply to the Review Report (Reviewer 2)
Please provide a point-by-point response to the reviewer’s comments and either enter it in the box below or upload it as a Word/PDF file. Please write down "Please see the attachment." in the box if you only upload an attachment. An example can be found here.
* Author's Notes to Reviewer
FileEditViewInsertFormatToolsTableHelp
Paragraph
P
0 WORDS
Word / PDF
or
Bottom of Form
Top of Form
Review Report Form
Open Review
(x) I would not like to sign my review report
( ) I would like to sign my review report
English language and style
( ) Extensive editing of English language and style required
( ) Moderate English changes required
( ) English language and style are fine/minor spell check required
(x) I don't feel qualified to judge about the English language and style
|
Yes |
Can be improved |
Must be improved |
Not applicable |
|
|
Does the introduction provide sufficient background and include all relevant references? |
( ) |
(x) |
( ) |
( ) |
|
Is the research design appropriate? |
( ) |
(x) |
( ) |
( ) |
|
Are the methods adequately described? |
( ) |
( ) |
(x) |
( ) |
|
Are the results clearly presented? |
( ) |
( ) |
(x) |
( ) |
|
Are the conclusions supported by the results? |
( ) |
(x) |
( ) |
( ) |
Comments and Suggestions for Authors
The manuscript "Snack food consumption across the Pune transect in India: A comparison across gender, age, education, socioeconomic levels, and locations” by Devesh et al. is quite interesting and covers a topic that was not explored in MDPI journals yet. The study was aimed to learn the snack food consumption, eating behaviors associated with snacking, and anthropometric measurements in a group of 1 400 Indian consumers.
In my opinion, authors should consider shorting the title of this manuscript; there is no point to write all social-demographical aspects measured (they are typical). Maybe you should make it more informative e.g. add information about anthropometrics, and eating behaviors. This is only a kind suggestion for authors, but the final decision is theirs.
Response- Many thanks for the suggestion. We have shortened the title to Snack food consumption across Pune transect: A comparison of dietary behaviors based on consumer characteristics and locations
I think that the authors do not use correct template of Nutrients’ MDPI Journal. Where are the affiliation details and the prospective publication details taken from the left margin? Thank you. We have now provided the details as per the Nutrients’ MDPI Journal template.
This manuscript is full of text editing issues, so it is very hard to focus on more methodological aspects. There are just examples, please check this manuscript thoroughly:
- L5: why “This study” is bolded, why the hyphen is inserted here?-Response- It is corrected
- L 12: “influence>Three-quarter” – what is “>” sign for?- that sign is removed
- L23: please remove this hyphen.-Done
- L42: often-low – please separate it and remove the hyphen.- Done
- L92: “(NHFS)” double bracket should be on the right side.- It is corrected
- L158: “WHO Refs” – probably authors wanted to insert here citation.- corrected
- L172, L175, L193, L197, and elsewhere as appropriate: “Error! Reference source not found” -There was problem due to cross referencing. Now those are corrected
- L210: “Note: ***,** stands for 1%, 5% significance level. Ns-not significant 0”. What is the “0” in the end for? What about one *?-“0” was a typo. It is corrected. We have given a detailed explanation now at the bottom of the table; one * is for 10% significance level
- L310: “Mithra et al (206)” What does 206 mean?- Sorry for the typo. It is supposed to be 2006
I do not feel qualified to correct the English language, but I am sure that some parts of the manuscript require and others would benefit from professional proofreading. Please look at some examples. The text needs to be checked thoroughly: Done
- L31-32: “Snack foods can….” – this sentence is not clear. Require grammar corrections.- We modified the writing for better composition and coherence
- L45-47: “US [10] and [11] for Australia assess the” – not clear. We modified the language
- L48: “Studies conducted on UK, shows a positive”, not clear, should be rather: “Studies conducted in the UK show…”- Corrected
- L49: I suggest inserting “as well as” between “SSB in children” and “negative association”- Done
- L58: “In roughly comparable contexts, a study, in West Java, Indonesia, study SFC” – not clear, at all.- We added comparable context in terms of SES of the study subjects as opposed to the rich countries like UK and Australia
- L64: “In the decade since …” – not clear…- We changed the text to the following. Since then, in a decade, there has been a significant food system transformation in India but the effects of which could not be analyzed due to data unavailability
- L76: “archetypal middle-income city” What does it mean?- We changed to typical or similar
- L94-96: “After seeking…”- not clear- We changed the text to be clearer
- 160: “At weekly frequency, in all, 90% had SFC” – incorrect. We had included tea, coffee (beverages) in the snacking, hence it was showing 90%. We have removed tea now. Instead of weekly we have reported only on monthly frequency now.
- L287: “Results here show” – this is not scientific language- We took out results here show. Thanks for suggesting the correct presentation
- L304: “Ordering outside food” Ordering takeaway? dining out? – not clear- It covers both outside food as well as take away food
- … and many others across the entire manuscript. At some point in this manuscript, the plethora of mistakes caused I stopped finding, correcting both editorial and language issues. This should be done by authors or professional services.- Many thanks for this suggestion. We internally got it checked for grammatical and presentation issues.
Introduction
I general, I think that introduction provides enough background, Nevertheless, please consider my suggestions below:
- L24-28: Please provide a reference- We have provided a reference
- L29: Please provide the full name of SFC in the main text.- We have provided the long form as snack food consumption
- L51: Please provide the full name of SES in the main text. It is a standard scientific practice to provide the full name of the acronym when used for the very first time.- we have provided the full form as socio economic status, the first time the term has been used
- L56: In my opinion, authors should expand this section a bit to provide the newest findings from others studies on Snack Food Consumption in India.
- Bhol A, Sanwalka N, Kapasi TA, Piplodwala SZ, Ansari LM, Katawala FM. Changes in Snacking Patterns during COVID-19 Lockdown in Adults from Mumbai City, India. Current Research in Nutrition and Food Science Journal. 2021 Sep 10;9(3).
- and maybe some others…
Response-We have added papers from recent periods on dietary behaviors in India
Materials and methods
- The subsection “study population and design” should begin with a brief description of study design, i.e. generic information on what was assessed and how many people participated in this study. Then the details can be described. Done
- When referring to SLI and NHFS appropriate references should be added. Done
- Maybe a chart will be more informative to illustrate to the readers how this study was performed. We have added a chart
- The methodology description is missing information on how exactly the interview was performed (did recruit participants started with completing the Food Frequency Questionnaire, then they completed a questionnaire on eating behavior, finally anthropometric measurements? How long does the interview take? Did participants complete the questionnaire by themselves or were answering to designated researchers). We have added the details now
- Each questionnaire description should indicate the number of questions asked. Done
- L103-110: Separate paragraph should be dedicated for statistical analyses. Usually is located at the end of the material and methods section. Done
- L80; L82 and elsewhere as appropriate: Sentences should never begin with numerals, I kindly advise paraphrasing all sentences that use such construction. We have corrected
Results
- The authors collected interesting data in a big sample of consumers, but the presentation manner should be corrected. The scope of an article could be easily split into two valuable articles. Currently, authors try to squeeze everything into one article and this makes it hard to understand and find the most important outcomes. Maybe a cluster analysis would help to find the patterns that the authors refer to in the abstract. Thank you for this valuable suggestion. In future research we will organize our paper based on cluster analysis.
- Table 1. The official symbol International System of Units (SI) for gram is "g" following the numeric value with a space. In my opinion, authors should check this table again is quite complicated and unnecessarily stretched. Removing of 1,5 line spacing in the second row of table 1 and changing “Pearson chi-square” for “χ2” should make the table shorter. We have made changes in the table.
- Authors present the Hunger (%) – where was that explained in methodology? We have now explained in the methodology in the “statistical analyses”
- In my opinion figure, 1 is not legible enough. Maybe authors should consider moving the legend from the left side under the set of circles and expanding them a little bit. We have done that now
- The title of the figure is usually located under the graphs. Authors should refer to the figure in the text, which is not the case. Done
- The subtitle“Statistical Analysis” should be renamed – the statistical analysis subtitle is usually located in the material and methods section. Statistics should be used for all data to evaluate the significance of the results, hence a specific subsection is pointless. We have now included “statistical analyses” as a subtitle in the material and methods section only.
- It is not acceptable to insert a table that is two pages long and does not provide an explanation for it. Moreover, only the most important results should be presented. The legend is incorrect. We have shortened the table and provided only variables of interest
- The authors collected anthropometric data but they did not show any of them and did not find any relationship between BMI and snacking, which is a known factor of obesity development. Thanks for pointing this out. We also find there is no relationship between BMI and snacking as found in other studies but contextually it might be different because of hunger snacking and physical activity.
Discussion
- In my opinion, the discussion is written moderately poorly, as barely tackles aspects that were somehow selected by authors. There is a lack of discussion on titular SCF in India. The probable reason for such a state is a too vast scope of the article. I am really curious about other reviewers' opinions. In the revision we have tried to streamline the discussion section.
Conclusions
- Generic, but I am wondering which of these conclusions are discoveries? We have shortened the conclusion and made it more specific
- No other comments for this section, as is nicely written
- SFC, ST – please provide here a full name. Done
Institutional Review Board Statement – authors can provide the number of resolution if is available-Number is not available. Please see below the Ethics Committee approval letter.
References
- In the manuscript, 50 references were used, but less than 25% of them were published within 5 past years. It shows the need to refresh the references a bit. There are many interesting articles, even from this publisher, which findings support your opinion. We have updated the references
Abstract
- I think the abstract should be also corrected. In my opinion, the first thing you should do is to provide 1-2 short pieces of information on the study context/ background, then what was the aim. After that, the most important measures should be briefly described. Does dietary data describe the physical activity and BMI, as stated here? Then you can report the most important results, but do not forget about some on snack food consumption, which is in the manuscript title. We have reworked the abstract as suggested. Many thanks for suggesting a proper structure for the abstract.
According to journal standards, my review categorizes this manuscript as requiring major revisions. I think the authors might need more time to improve the manuscript, but a chance to do it should be given.
Reviewer keeps fingers crossed and waits for the revised version.
Submission Date
25 October 2021
Date of this review
18 Nov 2021 11:59:46
Bottom of Form
© 1996-2021 MDPI (Basel, Switzerland) unless otherwise stated
Disclaimer Terms and Conditions Privacy Policy
Round 2
Reviewer 1 Report
The manuscript entitled ‘Snack food consumption across the Pune transect in India: A comparison of dietary behaviors based on consumer characteristics and locations’ (new tile) presents interesting issue. I appreciate the great efforts that the authors have made in response to my questions and concerns. However, the conclusion should be shorted.
Author Response
Comments and Suggestions for Authors
Reviewer 1
The manuscript entitled ‘Snack food consumption across the Pune transect in India: A comparison of dietary behaviors based on consumer characteristics and locations’ (new tile) presents interesting issue. I appreciate the great efforts that the authors have made in response to my questions and concerns. However, the conclusion should be shorted. We have shortened the conclusion section and put subsection for further research separately.
Reviewer 2
Comments and Suggestions for Authors
Dear Authors,
I really appreciate your effort. I read your manuscript again. Last time, I gave you some ideas on how to improve this manuscript, I see big progress, but some of the mentioned parts still require revision. Moreover, I noticed other aspects that should be revised.
Please check the manuscript for editorial and language mistakes again. Maybe it will be easier to see after you accept all changes made.
- I still do not see publication details that were deleted from the left margin. Please add it. We had made revisions in the last updated version. Submitted to Dr. Zang by email on 26th Nov 2021. The changes are other in the updated version. It is possible that the changes were missed in versions
- L26: “Dietary behaviors of slum dwellers was characterized by three-quarters of them having SFC with family” – this sentence has no logic point. How someone can have SFC (Snack food consumption) with family? This has been addressed in the document. We have changed the language to “together with family members at home.”. Here we are trying to reflect that majority of the slum dwellers were consuming snacks with their family members at home.
- L91-96: Please provide the reference. We have provided the source in the text.
- L108: In my opinion, authors should consider adding the full questionnaire used in this study as supplemental materials. The current description of the experiment does not allow another researcher to reproduce the results. This is not the total author's fault, because the manuscript volume is limited. Questionnaire has been submitted in the supplementary material.
- L136-138: Please check with an English linguist if an abbreviation like (10 to 12y) is correct. In my opinion, this is slang that should not be used in scientific papers. However, I am not a native speaker. We have made the required change in the updated paper. We have mentioned (10 to 12 years), same has been followed for other age groups.
- L190: “A variety of two- and three-dimensional food models (bowls, spoons, roti sizes)” – what do you mean by that: food or utensils? This has been addressed in the document as a footnote and information is also provided in the questionnaire in the supplementary material.
- L214- 215: Please provide reference For METs values. Reference has been provided.
- L245-247: Please provide the reference for WHO guidelines. Reference has been provided.
- L249: In which module consumers were asked about hunger status? In what way? There is no information on this topic in the material and methods section. In the paper, we have asked about reasons of snacking and not hunger status per as. Amongst the major reason of snacking was hunger. We have made the change in the updated paper.
- L250: “mean per capita SFC”? – is it correct? This has been corrected in the document. Another line “Mean snack consumption was 9.1 items per week” has been replaced as “average snack consumption was 9.1 items per week”.
- L248: “Statistical analyses” → Authors usually name this section in the singular form as Statistical Analysis. I kindly advise the authors to revise the first paragraph of this section to make the language plainer and clearer. You wrote “…This was done…” – what was done? In my opinion, you should start with the statistical software, then clearly state what is computed and how. We have made changes in the paper.
- L325 “36.6” – where is the unit? We have made changes in the paper.
- Table 1: I kindly suggest to precise the name of the “Hunger (%)“ column, it is very misleading. Remember to explain in the questionnaire description how was that measured. We have changes the “Hunger (%)”column as “Hunger as a reason of snacking, changed in the document.
- L331: “Weight wise SFC” – not clear. Scientific language needs to be precise and easy to understand for international readers. Did you mean that “the weight of consumed snack was the highest…”? We have changed the language in the paper.
- L357: “monotonic with SES” – what does it mean? We have changed the language in the paper.
- Figure 2. The authors need to provide better quality graphs I still cannot read the data. We have updated the figure for better visibility.
- Look at Figure 2, yellow brick in legend claims “(without tea) Tea” – what is the sense of it. What are packed snack? Are they something different from sweet and savory snacks? Some of it packed or some of it is not. We have corrected the legend in figure 2.
- Table 3. Why do you present the table 3, there is no description of table content neither before nor after the table. What is the point to do it? Every single table and figure should be referenced in the text (not in the material and methods section) and at least briefly described. We have provided the description in the results section.
- L438: “4. 4Discussion” – double 4. We have corrected this it in the paper.
- L539-543: “Author Contributions” – In my opinion, only initials should be provided here. This had been updated in the last version sent. The new version has this change.
Reviewer 2 Report
Dear Authors,
I really appreciate your effort. I read your manuscript again. Last time, I gave you some ideas on how to improve this manuscript, I see big progress, but some of the mentioned parts still require revision. Moreover, I noticed other aspects that should be revised.
Please check the manuscript for editorial and language mistakes again. Maybe it will be easier to see after you accept all changes made.
- I still do not see publication details that were deleted from the left margin. Please add it.
- L26: “Dietary behaviors of slum dwellers was characterized by three-quarters of them having SFC with family” – this sentence has no logic point. How someone can have SFC (Snack food consumption) with family?
- L91-96: Please provide the reference.
- L108: In my opinion, authors should consider adding the full questionnaire used in this study as supplemental materials. The current description of the experiment does not allow another researcher to reproduce the results. This is not the total author's fault, because the manuscript volume is limited.
- L136-138: Please check with an English linguist if an abbreviation like (10 to 12y) is correct. In my opinion, this is slang that should not be used in scientific papers. However, I am not a native speaker.
- L190: “A variety of two- and three-dimensional food models (bowls, spoons, roti sizes)” – what do you mean by that: food or utensils?
- L214- 215: Please provide reference For METs values.
- L245-247: Please provide the reference for WHO guidelines
- L249: In which module consumers were asked about hunger status? In what way? There is no information on this topic in the material and methods section.
- L250: “mean per capita SFC”? – is it correct?
- L248: “Statistical analyses” → Authors usually name this section in the singular form as Statistical Analysis. I kindly advise the authors to revise the first paragraph of this section to make the language plainer and clearer. You wrote “…This was done…” – what was done? In my opinion, you should start with the statistical software, then clearly state what is computed and how.
- L325 “36.6” – where is the unit?
- Table 1: I kindly suggest to precise the name of the “Hunger (%)“ column, it is very misleading. Remember to explain in the questionnaire description how was that measured.
- L331: “Weight wise SFC” – not clear. Scientific language needs to be precise and easy to understand for international readers. Did you mean that “the weight of consumed snack was the highest…”?
- L357: “monotonic with SES” – what does it mean?
- Figure 2. The authors need to provide better quality graphs I still cannot read the data.
- Look at Figure 2, yellow brick in legend claims “(without tea) Tea” – what is the sense of it. What are packed snack? Are they something different from sweet and savory snacks?
- Table 3. Why do you present the table 3, there is no description of table content neither before nor after the table. What is the point to do it? Every single table and figure should be referenced in the text (not in the material and methods section) and at least briefly described.
- L438: “4. 4Discussion” – double 4.
- L539-543: “Author Contributions” – In my opinion, only initials should be provided here.
Author Response

(The authors gave the same response as above.)
